# A Senescent Cluster in Aged Human Hematopoietic Stem Cell Compartment as Target for Senotherapy

**DOI:** 10.3390/ijms26020787

**Published:** 2025-01-17

**Authors:** Laura Poisa-Beiro, Jonathan J. M. Landry, Bowen Yan, Michael Kardorff, Volker Eckstein, Laura Villacorta, Peter H. Krammer, Judith Zaugg, Anne-Claude Gavin, Vladimir Benes, Daohong Zhou, Simon Raffel, Anthony D. Ho

**Affiliations:** 1Department of Medicine V, Heidelberg University, 69117 Heidelberg, Germany; laura.poisa.beiro@gmail.com (L.P.-B.); michael.kardorff@med.uni-heidelberg.de (M.K.); v.eckstein@t-online.de (V.E.); simon.raffel@med.uni-heidelberg.de (S.R.); 2Molecular Medicine Partnership Unit, European Molecular Biology Laboratories (EMBL) & Heidelberg University, 69117 Heidelberg, Germany; 3Genomics Core Facility, European Molecular Biology Laboratories (EMBL), 69117 Heidelberg, Germany; landry.jo@gmail.com (J.J.M.L.); laura.villacorta@embl.de (L.V.); benes@embl.de (V.B.); 4Department of Pharmacology and Therapeutics, University of Florida, Gainesville, FL 32611, USA; yanbowen@ufl.edu; 5German Cancer Research Center, 69120 Heidelberg, Germany; peter.h.krammer@gmail.com; 6European Molecular Biology Laboratories (EMBL), 69117 Heidelberg, Germany; judith.zaugg@embl.de; 7Department of Cell Physiology and Metabolism, University of Geneva, 1205 Geneva, Switzerland; anne-claude.gavin@unige.ch; 8Diabetes Center, Faculty of Medicine, University of Geneva, 1205 Geneva, Switzerland; 9Department of Biochemistry and Structural Biology, University of Texas Health San Antonio, San Antonio, TX 78229, USA; zhoud@uthscsa.edu

**Keywords:** hematopoietic stem and progenitor cells (HSPC), aging, senescence signature, comparative single-cell transcriptomics

## Abstract

To identify the differences between aged and young human hematopoiesis, we performed a direct comparison of aged and young human hematopoietic stem and progenitor cells (HSPCs). Alterations in transcriptome profiles upon aging between humans and mice were then compared. Human specimens consist of CD34+ cells from bone marrow, and mouse specimens of hematopoietic stem cells (HSCs; Lin− Kit+ Sca1+ CD150+). Single-cell transcriptomic studies, functional clustering, and developmental trajectory analyses were performed. A significant increase in multipotent progenitor 2A (MPP2A) cluster is found in the early HSC trajectory in old human subjects. This cluster is enriched in senescence signatures (increased telomere attrition, DNA damage, activation of P53 pathway). In mouse models, the accumulation of an analogous subset was confirmed in the aged LT-HSC population. Elimination of this subset has been shown to rejuvenate hematopoiesis in mice. A significant activation of the P53–P21WAF1/CIP1 pathway was found in the MPP2A population in humans. In contrast, the senescent HSCs in mice are characterized by activation of the p16Ink4a pathway. Aging in the human HSC compartment is mainly caused by the clonal evolution and accumulation of a senescent cell cluster. A population with a similar senescence signature in the aged LT-HSCs was confirmed in the murine aging model. Clearance of this senescent population with senotherapy in humans is feasible and potentially beneficial.

## 1. Introduction

With age, the hematopoietic stem and progenitor cells (HSPCs) undergo quantitative and functional changes [1]. In animal models, the number of primitive HSPCs in the bone marrow might increase with age but these HSPCs exhibit limited regenerative potential [2,3,4,5,6]. The functional decline can probably be extrapolated to the human system as HSPCs harvested from elderly donors are associated with a decreased transplantation success rate [7].

The reduced engraftment potential is a consequence of interactions between HSPCs and the cellular determinants in the bone marrow niche [8,9]. The latter includes mesenchymal stromal cells (MSC), endothelial cells, as well as macrophages/monocytes [10,11,12,13]. Transplantation of aged HSPCs into young mice was not able to improve regenerative function [2,6,14]. Previous studies from our group have provided evidence that the intrinsic property of human HSPCs contributes significantly to the aging process [15,16,17].

Whether the functional decline upon aging occurs globally in the HSPCs, or is confined to a specific subpopulation within the HSPCs has remained controversial [1,18,19]. These two mechanisms may not be mutually exclusive. As HSPCs experience damage in their microenvironment, some of them undergo senescence [20]. The accumulation of senescent cells can further affect the remaining functional stem cells, creating a feedback loop that exacerbates the decline.

While aging is defined as the progressive changes that lead to functional decline, senescence is a stable form of cell cycle arrest induced by damage in individual cells [21,22,23,24,25,26]. Stress induces senescence by causing DNA damage and telomere attrition [27,28,29]. The purpose of senescence is to limit the proliferation of damaged, genomic instable cells. Senescence also plays a physiological role during normal development, e.g., during embryogenesis, senescence of specific cells is needed for tissue homeostasis [30]. Within the aging process, factors that initiate cellular senescence include telomere attrition and DNA damage. These damages subsequently lead to cell cycle arrest and activation of anti-apoptotic factors [27,28,31,32]. In animal models of aging, accumulation of senescent cells in the hematopoietic stem cell (HSC) compartment might lead ultimately to an overall decline in regenerative potential [22,23,33].

Transcriptome studies in murine HSPCs revealed that the senescence phenotype is characterized by DNA damage response, chromatin remodeling, and cell cycle arrest [34,35,36,37,38,39,40,41,42,43]. In animal models, interventions targeting senescent cells induced an improvement in aging disorders [22,23,25,26,33]. While there are encouraging preclinical examples of successful treatment strategies that target senescent markers [44], we have yet to define the appropriate targets. Albeit a large body of literature has focused on senescence-associated inflammatory phenotype (SASP) as targets for senotherapeutics, SASP may represent merely a downstream mechanism.

The goal of this study is to determine if there is a global shift in the human HSC compartment, or if a specific subset of population within the HSPC compartment that drives the aging process. Another important aim is the identification of markers and pathways that define these senescent cells in human hematopoiesis as compared to those found in mouse models.

## 2. Results

### 2.1. GSEA Between Old and Young Subjects in HSPCs

A cohort of 12 human subjects were categorized according to age into two classes: old (≥59 years, *n* = 7, median = 67 years), or young (≤31 years, *n* = 5, median = 30 years). Detailed features are summarized in Appendix A. After quality control, normalization, principal components analysis and data projections using UMAP, the scRNA-seq dataset of the 12 human samples contained 34,408 native features [45,46,47,48,49]. Upon collapsing features into gene symbols, there were 24,299 genes identified.

The most significant findings in Gene-Set Enrichment Analysis (GSEA) [50,51] comparing the whole HSPCs derived from old versus young human subjects are summarized in Figure 1. Analysis of “Reactomes” [52], for example, shows that 138 of the 1025 gene sets are enriched for old CD34+ cells at a *p* value of <0.01. Prominent alterations include “Telomere damage stress induced senescence”, “Formation of senescence associated heterochromatin foci (SAHF)”, “TNFR1 induced NFKB signaling pathway”, “DNA methylation”, “Senescence associated secretory phenotype (SASP)”, “Glycogen metabolism”, “Inhibition of DNA recombination at telomere”. 69 of the 1025 gene sets are enriched in young CD34+ cells and the pathways “Cell cycle” and “DNA repair” were most prominent.

A selection of enrichment plots on specific pathways or biological processes from Reactomes, KEGG, and Hallmarks are depicted in detail on the right [52,53,54]. Significant enrichments were found in old subjects for “DNA damage telomere stress induced senescence”, “Formation of senescence associated heterochromatin foci (SAHF)”, “TNFR1 induced NFKB signaling pathway”, “DNA methylation”, “Senescence associated secretory phenotype (SASP)”, “Glycogen metabolism”, “Inhibition of DNA recombination at telomere”. Gene sets involved in “Cell cycle” and “DNA repair” were enriched in the young subjects. (Color of dots = *p*-values; Size of dots = Enrichment Score.)

### 2.2. Developmental Trajectories of HSPCs in Old and Young Subjects

The structure, nodes, branches, and the composition of the 14 clusters are mapped in Figure 2A. Starting with the root (white circle 1 in Figure 2A), the development from HSC to (a) myeloid-lymphoid progenitors (MLP), (b) megakaryocyte and erythrocyte progenitors (MEP), (c) granulocyte–monocyte progenitors (GMP), (d) granulocytic precursors (G), monocytic precursors (Mono) and dendritic cell (DC) precursors, and to various subsets of B lymphocyte precursors (Ly1 to Ly4) is illustrated. There is an early split from HSC that separates MEP from MLP. The latter diversifies into granulocyte–monocyte progenitors (GMP) and lymphoid precursors (Ly1 to Ly4).

The most prominent differences between old and young humans are: an increase in a subcluster of HSCs, and accumulation of a subpopulation of lymphoid precursors (Ly4) in old subjects (Figure 2B,C). While a significant increase in HSCs, G-, DC- precursors and in Ly4 is found in old subjects, higher percentages of MLP, MEP, GMP, Mono, Ly2 and Ly2A are found in young subjects (Table 1). In this study, we focus on the subclusters in aged HSCs, and especially on the differences in these subclusters between aged and young human subjects.

Figure 2A depicts the developmental trajectory and clustering of HSPCs along the course of hematopoietic differentiation pathway. The black lines show the structure of the trajectory. The circles with numbers denote special points within the trajectory.

The relative developmental times (pseudotimes) of the respective clusters from the old versus young subjects are shown in Figure 2D,E. Statistical analysis with the Kolmogorov–Smirnov test [55] showed a remarkable delay in pseudotime of the HSC development in old subjects (*p* < 0.001). This observation supports the notion that, albeit the proportion of HSCs increases with age, their developmental process is significantly delayed.

### 2.3. GSEA—Comparisons in the Developmental Stages Within the HSPC Compartment

We initially performed GSEA between old and young subjects using the aging signature gene sets as reported by Kirschner et al. [37], Svendsen et al. [56], and Saul et al. [57]. Upon reviewing the leading-edge subsets, the top genes from the respective gene sets were selected and synthesized into the “Aging signature human HSC”. We identified 40 genes that were up-regulated with age, and 7 genes that were down-regulated (Appendix A).

We then analysed the GSEA differences between the two age groups in the various differentiation stages. Comparisons between old and young subjects were performed in the followings: (a) HSC, (b) MLP, (c) MEP, and (d) GMP. Pathways that are significantly increased in the HSC and MLP compartments in old subjects include: “Aging signature human HSC”, “Formation of SAHF”, “DNA damage telomere-stress induced senescence” (Figure 3). While a significant increase in “TNFR1 induced NFKB signaling pathway”, “Cellular response to hypoxia”, “P53 pathway”, was found only in the HSCs, elevation of “Senescence associated secretory phenotype” was found further downstream in the MLP and GMP compartments and “DNA methylation” in the GMP.

### 2.4. Analyses of the HSC Subclusters

As the number of primitive HSCs from each individual was extraordinarily rare, we recruited additional human subjects (*n* = 3; 2 old and 1 young, Appendix A) to study this subset in more depth. Using a higher resolution (k = 15, resolution = 1 × 10^−2^) for clustering of HSCs (*n* = 578 cells from 15 subjects), we were able to separate this population into 6 subclusters according to their hierarchical order (Figure 4A). We assigned these sub-clusters: multipotent progenitor 1 (MPP1), MPP2, MPP2A, MPP3, preMLP and preMEP, respectively. The most remarkable difference between the two age groups was found in the MPP2A subcluster (Figure 4B), which comprised 22.7% of HSCs in old subjects, versus 9.4% in young subjects (*p* < 0.001, Table 2). Remarkable is also that the MPP2A subcluster side-tracks from the main differentiation trajectory.

The changes in differential expression of selected genes within the gene set “Aging signature human HSC” in the 6 subclusters within the HSC compartment were interrogated. Prominent differences were found in the MPP2A subcluster (Figure 4D). The expression levels of ATF3, CDKN1A, CLU, FOSB, ID1, JUN, MAFF, NR4A2, PLK3 were elevated in old subjects. The statistical significance of these alterations is confirmed in volcano plot (Figure 4F). The magnitude of difference was greatest in CDKN1A and ATF3, MAFF, NR4A2, PLK3, FOSB and ID1, while the expressions of POT1 and TERF1 were decreased. Cell cycle analysis also showed that the MPP2A subcluster in old subjects was in G_0_ cell cycle phase and compatible with the notion that these cells were in cell cycle arrest.

We then examined the differential expression comparing MPP2A subcluster with all the other subclusters within the HSC compartment (Figure 4E). In the MPP2A subcluster, the genes: NR4A2, MAFF, JUN, ID1, FOSB, CLU, CDKN1A and ATF3 were again increased. The significance of these changes was further confirmed by volcano plots.

### 2.5. Aging Transcriptome Profiles—Comparison Between Human and Mouse Data

We reviewed all other changes in gene expressions that were significantly altered by more than twofold but were not able to discover any surface marker genes among them. Isolation of the senescent cells by surface marker for functional assays is hence not possible. In a murine model, it has been shown that a reduction in senescent cells in the HSC compartment was able to rejuvenate the hematopoietic system [33]. The same group (B.Y. and D.Z. among the authors of this article) has in the meantime performed single-cell transcriptome studies on HSCs in aged (100 weeks) and young (8 weeks) mice. For validation of the senescent MPP2A subcluster, we have compared our human data with the alterations in transcriptome profiles comparing the aged versus young LT-HSCs from this mouse aging model. Of the 14 selected genes prominently increased in the human MPP2A subcluster, 10 could be identified in the aged LT-HSC of the animals while 4 were beneath detection levels. Six genes are significantly elevated: Atf3, Clu, Fosb, Id1, Jun, Plk3, while the differences in Cdkn1a, Maff, Notch1 and Stat3 are statistically not significant and Nr4a1 is up-regulated in mice (Figure 5A). With the exception of Nr4a1, the expression pattern illustrated in Figure 5A in LT-HSC from aged versus young mice is similar to that in human MPP2A subcluster shown in Figure 4D.

### 2.6. Identification of Senescent Population in the Primitive HSC Compartment in Other Human Bone Marrow Datasets

Applying our algorithm, we have analyzed the developmental trajectory of the primitive HSC compartment in other human bone marrow datasets published in the literature (Zhang et al., 2022 [58]; Oetjen KA et al., 2018 [59]). In addition to the identification of an MPP2A analogous subset from the old age group, an almost identical elevation of the senescence genes (ATF3, CDKN1A, CLU, FOSB, ID1, JUN, MAFF, NR4A2, PLK3) was found in the dataset from Zhang et al. (Figure 5B) and a similar pattern in that from Oetjen et al. (Figure 5C).

A remarkable finding throughout our analysis is a significant up-regulation of CDKN1A in aged HSC, especially in the early stages of development, i.e., in MPP1, MPP2, MPP2A (Appendix A). Simultaneously, GSEA has established a highly significant activation of P53 pathway in aged HSC (Figure 3). The onset of senescent state has been shown to involve one of the two major tumor suppressive pathways, P53–P21 and p16 Ink4a-pRb in animal models [60,61,62]. The P53–P21 pathway is closely linked to elevated expressions CDKN1A, and the P16 INK4A pathway to elevated CDKN2A and CDKN2B. In mice, significantly increased expressions of Cdkn2a and Cdkn2b represent hallmarks of senescence [60,61,62]. Both genes are, however, expressed in very low to non-detectable levels in human HSCs. Our results have provided clear evidence that, in contrast to murine models, an activated P53–P21 pathway is the main mechanism for aging of the HSC compartment in human subjects.

We have also performed GSEA using several gene sets that might identify aging signatures in human HSPCs [37,56,57]. As summarized in Appendix A, these gene sets were not able to consistently identify senescent cells in human HSPCs.

## 3. Discussion

The most remarkable and novel findings are the accumulations of two unique clusters in the bone marrow HSPCs of old human subjects. In the early phase of hematopoietic development, a subcluster “MPP2A” in the HSC compartment is significantly increased in old subjects. The transcriptome profile of this subcluster reveals that these cells are enriched in HSCs with senescence signatures. We searched for all other significant changes in gene expressions in HSCs between the two age groups and were not able to discover any surface marker genes that would have rendered an isolation of these senescent cells for functional assays possible. The accumulation of a lymphoid precursor subset Ly4 further downstream along the differentiation pathway is also remarkable and requires additional functional studies. The latter are still work in progress and beyond the scope of the present manuscript.

In a murine model, the group of D. Zhou has demonstrated that reduction in senescent cells in the HSC compartment was able to rejuvenate the hematopoietic system [33]. Single-cell transcriptome data comparing the LT-HSCs between aged and young mice from this model have now been generated. This has enabled us to compare the alterations in transcriptome profile upon aging between the two species. We have demonstrated an accumulation of cells with analogous senescence signatures from human study in the aged LT-HSC compartment in this mouse model. Clearance of this population, characterized by Cdkn2a and p16Ink4a expression in mice, either genetically or by inhibitors of anti-apoptotic factors, has been shown to rejuvenate the hematopoietic system in aged animals [22,23,33].

A comparison of the transcriptomes from the MPP2A subcluster versus the other subclusters within the HSC compartment in humans again confirmed the enrichment of senescent cells in this subset. Applying our algorithm, we were able to identify an analogous senescent population in the aged HSC compartment of human BM datasets from other authors [58,59].

In mouse models of aging, telomere attrition and DNA damage result in induction of one of two major tumor suppressive pathways, P53–P21 and p16^Ink4a^-pRb [60,61]. The activated Ink4a/Arf locus leads subsequently to sustained p53 activation and increased expressions of Cdkn2a and Cdkn2b [22,23]. Clearance of p16^Ink4a^ positive cells increases healthy lifespan in mice [23]. The expression levels of CDKN2A and CDKN2B in human samples have been found to be very low or non-detectable, independent of age. In contrast, we have established a consistently significant increase in expression of CDKN1A in aged HSCs and in MPP2A subcluster. Senescence in the human HSC compartment is probably coupled with P53–P21 and activated CDKN1A and not with P16ARF, as suggested by other authors [25].

The production of a combination of inflammatory factors named SASP has been regarded as a prominent hallmark for senescence in animal models [24,26]. A large body of literature has also focused on SASP as targets for senotherapeutics. We have demonstrated that activation of this pathway was not significant in the aged HSCs, nor in the MPP2A subcluster, but only further downstream in more differentiated stages such as LMP and GMP. Previous studies of BM samples from aged human subjects have often been harvested from hip replacement operations [50,58,59]. Such samples were derived from an inflammatory environment. The BM samples in this study were all harvested from the posterior iliac crest of healthy subjects according to a standardized protocol. Our data indicate that elevation of SASP does not represent the dominant mechanism for aging in the primitive HSC compartment, but more downstream in progenitors of myeloid differentiation. This may be of relevance in the design of future senotherapeutic strategies.

In summary, our study indicates that aging in the primitive HSC compartment is mainly caused by accumulation of a senescent subcluster. The latter is characterized by increased telomere attrition, activation of P53 pathway, cell cycle arrest, and a remarkable up-regulation of CDKN1A. Pathways such as activated SASP or dysregulation of DNA methylation represent more downstream mechanisms behind the aging mechanisms. Our data therefore favor the hypothesis that aging is not caused by a global shift in the HSC but by the accumulation of a senescent subset. The identification of this senescent subset in the human HSC compartment, in analogy to a similar population in LT-HSC in mice, indicates that clearance of this population with senotherapy is feasible and potentially beneficial.

## 4. Materials and Methods

Human specimens. Bone marrow samples were harvested from healthy human subjects through puncture at the posterior iliac crest. This study has been approved by the Ethics Committee for Human Subjects, University of Heidelberg, and written informed consent was obtained from each individual. The initial cohort of 12 human subjects were categorized according to age into two classes: old (≥59 years, *n* = 7, median = 67 years), or young (≤31 years, *n* = 5, median = 30 years). Detailed features are summarized in Appendix A. Preparation of human specimens were published previously [15,16]. The CD34+ cells were isolated using a FACS Aria II flow cytometry cell sorter (BD Biosciences, Franklin Lakes, NJ, USA).

Mouse specimens. Bone marrow cells were harvested from both young (6 to 8 weeks old) and aged (100 weeks old) mice, with 10 mice in each group. Hematopoietic stem cells (HSCs) were isolated via flow cytometry sorting (characterized as Lin− Kit+ Sca1+ CD150+ cells). Annotating the clusters was based on expressed genes previously reported [46].

Single-cell RNA-sequencing of HSPCs. Sequencing libraries from the human HSPCs were generated based on the smart-seq2 protocol of Picelli et al. [45] and the tagmentation procedure of Hennig et al. [47]. The details were published [15,16].

Data processing. Single-cell data preprocessing was performed using the programming language R. Raw reads were aligned using STAR aligner version 2.6.0a [48]. The count matrix generated for individual genes across cells in each sample was then subjected to further processing using the Bioconductor package Seurat V4 [49].

Gene-Set Enrichment Analysis (GSEA). We applied the GSEA (version 4.3.2., Broad Institute, Inc., Cambridge, MA, USA, Massachusetts Institute of Technology) for interpreting gene expression data [50,51]. We searched for significant differences in the gene sets defined in Hallmarks, Reactomes, KEGG, and Wikipathways [52,53,54].

Generating the gene set “Aging signature human HSC”. Using the gene sets for aging or senescence signatures in hematopoiesis as reported by Kirschner et al. [37], Svendsen et al. [56], and Saul et al. [57], we developed our unique gene set for human HSPCs (described under Results).

Toolkit for analyzing single-cell transcriptome data. We apply the toolkit Monocle 3 package within the R (version 4.2.2) environment for analyzing single-cell gene expression data [63,64,65]. Dimension reduction was based on uniform manifold approximation and projection (UMAP) [66], followed by clustering of cells. The workflow consists of organizing cells into trajectories, followed by statistical tests to identify genes that vary in expression over those trajectories.

Gene sets used for annotation of cell clusters. The gene sets used to classify and to define the lineage-specific signatures for the HSPCs are listed in Appendix A.

Cell cycle annotation. Each cell was scored for S phase and G2 phase using the marker gene list for the respective phase (Seurat package version 4.0.3), extracted from Tirosh et al. [67]). Using these 2 scores, each cell was then assigned to to G0, G1, S and G2-M according to Kowalczyk et al. [36].

Data availability. Raw data for the single-cell RNA-seq of human specimens have been deposited in the European Nucleotide Archive (ENA) database under accession ID PRJEB68076. Data for the mouse experiments are not yet publicly available. A manuscript with focus on dynamics of HSCs during aging and senolytic treatment is being prepared. Requests for original mouse data should be addressed to yanbowen@ufl.edu. The authors declare that all data supporting the findings of this study are available upon reasonable request.

## 5. Conclusions

Aging in the human HSC compartment is mainly caused by the clonal evolution and accumulation of a senescent cell cluster MPP2A. The latter is characterized by increased telomere attrition, cell cycle arrest, and a remarkable up-regulation of CDKN1A. The identification of a senescent subset in human non-primed HSC, in analogy to a similar population in LT-HSC in mice, indicates that clearance of this senescent population is feasible and potentially beneficial. For humans, the P53–P21CIP1 positive cells, and not the p16-Ink4a positive cells as in mouse models, represent the main target.

## Figures and Tables

**Figure 1 ijms-26-00787-f001:**
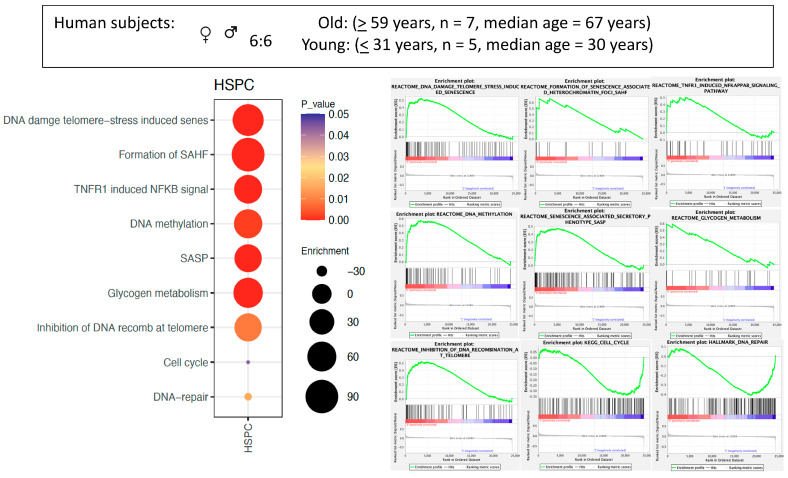
Gene-Set Enrichment Analysis (GSEA) comparing HSPCs (CD34+ cells) from the bone marrow of old versus young human subjects.

**Figure 2 ijms-26-00787-f002:**
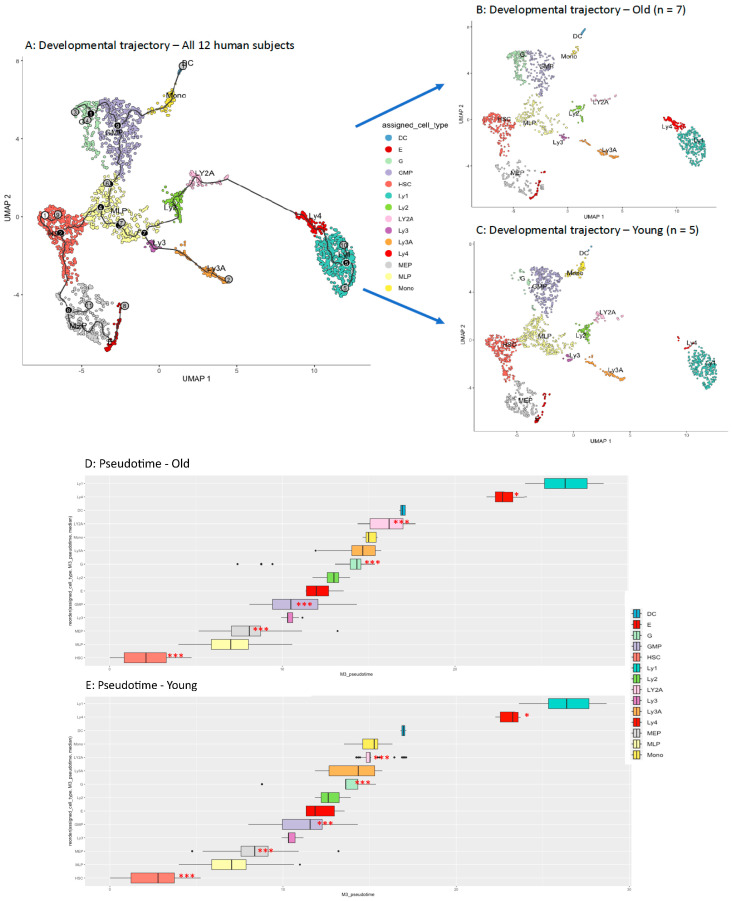
Developmental trajectory of human HSPCs. Starting with the root (white circle 1 in (**A**)), the development from primitive the HSC compartment to (i) myeloid-lymphoid progenitors (MLP), (ii) megakaryocte and erythrocyte progenitors (MEP), (iii) granulocyte (G)–monocyte (Mono) progenitors (GMP), and (iv) precursors for G, M and dendritic cells (DC), as well as to (v) various precursors of B lymphocytes (Ly1 to Ly4). The black lines show the structure of the graph. Each leaf, denoted by light gray circles, corresponds to a different outcome (i.e. cell fate) of the trajectory. Black circles indicate branch nodes, in which cells can travel to one of several outcomes. Numbers within the circles are provided for reference purposes only. (**B**) shows the developmental stages of HSPCs in old subjects, and (**C**) the developmental stages of HSPCs in young subjects. (**D**,**E**) illustrate the precise “pseudotimes” of each of the cell clusters and the results of statistical analysis in the old compared to young human subjects. The Kolmogorov–Smirnov test showed a significant delay in the median “pseudotimes” of the HSC, MEP, GMP, and in “Ly4” clusters in old subjects (box plots with *** = *p*-values of <0.001, box plot with * = *p* < 0.05).

**Figure 3 ijms-26-00787-f003:**
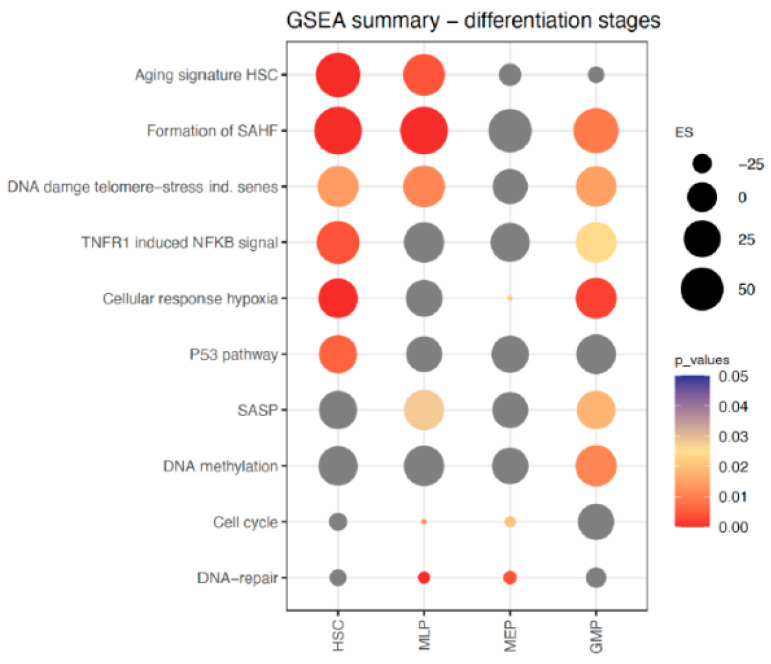
GSEA of expression profiles in pathways associated with senescence—comparisons between old and young subjects in early clusters of HSPCs. The pathways relevant for aging and senescence are listed on the y-axis. Comparisons between old and young subjects were performed in the following clusters as listed on the x-axis: (a) in HSC, (b) in MLP, (c) in MEP, and (d) in GMP compartments. (Color scale = *p* values, grey dots = *p* > 0.05, not significant; Size of dots = Enrichment Score).

**Figure 4 ijms-26-00787-f004:**
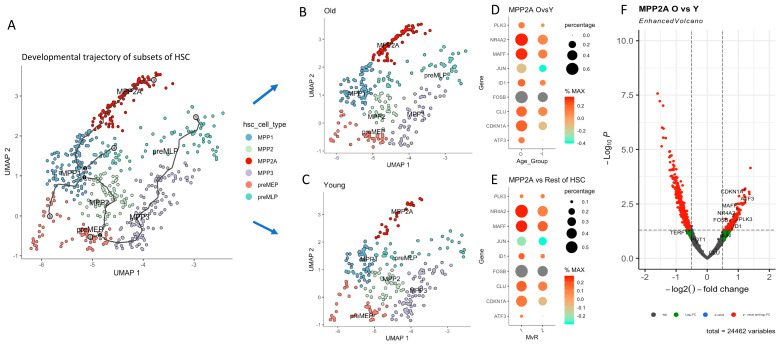
Subclusters of human HSCs and genetic signatures of MPP2A in old versus young subjects. (**A**) shows the developmental trajectory of the 6 subsets of HSCs in all human subjects; The black lines show the structure of the graph. Each leaf, denoted by light gray circles, corresponds to a different outcome (i.e. cell fate) of the trajectory. Black circles indicate branch nodes, in which cells can travel to one of several outcomes. Numbers within the circles are provided for reference purposes only. (**B**) shows the trajectory in old and in (**C**) the young subjects. In old subjects, a highly significant abundance in MPP2A was found. In (**D**) the differential expressions of selected genes between aged MPP2A versus young MPP2A are depicted, whereas in (**E**) the differential expressions between the MPP2A subcluster as compared to all other subclusters within the HSC population are shown. (x-axis MvR = MPP2A versus Rest of HSCs, 1 = MPP2A subcluster, 2 = other HSCs). The volcano plot (**F**) shows the magnitude and the significance of the differential expressions of senescence genes between old and young.

**Figure 5 ijms-26-00787-f005:**
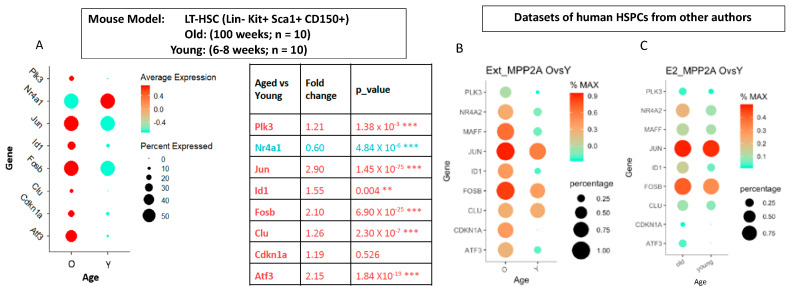
Validation of senescence gene signatures in mouse models and in other human datasets. (**A**) shows the differential expression of senescence genes in aged murine HSCs. By comparing the gene expression profiles of aged versus young LT-HSC, 6 of the most prominently elevated genes identified in human study were significantly elevated, while the increase in Cdkn1a was statistically not significant and Nr4a1 was elevated in young LT-HSCs. (Color of dots = average expression; Size of dots = percent of the cells expressed. The statistical analysis with Chi-squared test is shown in table in detail, ** = <0.01, *** = <0.001). (**B**,**C**) demonstrates that this senescence signature can be validated in human HSPC datasets from other authors. (Color of dots = relative expression; Size of dots = percent of the cells expressed).

**Table 1 ijms-26-00787-t001:** Composition of the HSPC clusters in all 13 subjects, in the old and in the young human subjects.

HSPC Clusters	All *n* = 3321 %	Old *n* = 1455 %	Young *n* = 1866 %	*p* Value
HSC	15.4	16.0	15.1	0.034 *
MLP	14.2	10.3	17.3	1.8 × 10^−15^ ***
MEP	10.0	8.5	11.2	4.0 × 10^−6^ ***
GMP	15.0	10.4	18.7	1.2 × 10^−18^ ***
G	6.5	12.5	1.9	1.9 × 10^−23^ ***
Mono	2.5	0.8	3.8	9.4 × 10^−11^ ***
E	2.5	2.4	2.6	0.127
DC	1.3	2.8	0.1	4.5 × 10^−9^ ***
Ly1	17.1	19.2	15.5	0.706
Ly2	3.4	2.8	3.9	2.5 × 10^−3^ ***
Ly2A	2.9	1.4	4.0	1.7 × 10^−8^ ***
Ly3	1.2	1.2	1.2	0.527
Ly3A	4.1	3.8	4.4	0.021 *
Ly4	3.8	8.0	0.5	3.6 × 10^−21^ ***

(Statistical analysis: * = <0.05, *** = <0.001, Chi-squared test).

**Table 2 ijms-26-00787-t002:** Composition of non-primed HSC sub-clusters upon higher resolution of clustering.

HSC Subclusters	All *n* = 578 %	Old *n* = 313 %	Young *n* = 265 %	*p* Value
MPP1	22.3	23.0	22.3	0.328
MPP2	13.0	11.2	15.1	0.564
MPP2A	16.6	22.7	9.4	2.7 × 10^−6^ ***
MPP3	20.2	15.3	26.0	0.052
PreMLP	13.7	13.1	14.3	0.735
PreMEP	14.2	15.3	12.8	0.122

(Statistical analysis: *** = <0.001, Chi-squared test).

## Data Availability

Raw data for the single-cell RNA-seq of human specimens have been deposited in the European Nucleotide Archive (ENA) database under accession ID PRJEB68076. Data for the mouse experiments are not yet publicly available as a manuscript with focus on dynamics of HSCs during aging and senolytic treatment is being prepared. Requests for original mouse data should be addressed to yanbowen@ufl.edu. The authors declare that all data supporting the findings of this study are available upon reasonable request.

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
