# Peer review of "A Senescent Cluster in Aged Human Hematopoietic Stem Cell Compartment as Target for Senotherapy"

_ijms, 2025, doi:10.3390/ijms26020787_

Round 1
Reviewer 1 Report
Comments and Suggestions for Authors
Poisa-Beiro et. al. utilized young and old human CD34+ single-cell sequence analysis and showed MPP2A fraction and Ly4 were significantly different featued fractions for aged hematopoiesis. MPP2A was senescent and had no differentiation by trajectory analysis, with increasing CDKN1A. The author also compared their results with mouse CD150+LSK data and other researchers’ data and showed partially overlapped gene expression signatures with them.
Major points
In the abstract, the author wrote, “To identify the contribution of senescent cells to aging of human hematopoiesis, we performed a direct comparison of aged and young human hematopoietic stem and progenitor cells (HSPCs).“ But “To identify the differences between aged and young hematopoiesis we performed a direct comparison of aged and young human hematopoietic stem and progenitor cells (HSPCs).“ and the result would be “described the contribution of senescent cells”.
line34, the author wrote, “A similar population in LT-HSC in mice could be validated.” there is no evidence of subpopulation imaged in mice data. The author should erase the sentence.
Fig. 1 (Paragraph 2.1) had no results. Please conclude the young vs aged GSEA analysis at the end of the paragraph.
In paragraph 2.3, they utilized 14 clusters and named each, but it wasn’t easy to follow considering the next paragraph, 2.5, of 6 clusters. How about changing the name of 2.3 (14 clusters) of HSC to HSPC? It was wondered why the subcluster name was MPP2A and Ly2A, Ly3A. Each expression pattern was similar to the main cluster? Please add the explanation in the manuscript.
Fig. 2 is too small to follow the data. please show much larger figures.
line263, “The P53-P21 pathway is closely linked to elevated expressions CDKN1A, and the P16 INK4A pathway to elevated CDKN2A and CDKN2B.”, line267 activated P53-P21 pathway and an elevated expression of CDKN1A”, line308”P53-P21 and activated CDKN1A”would be just redundant expression because CDKN1A is p21 and CDKN2A is p16. Rephrase or delete the sentence for the reader to easily follow.
Minor points
line154, in Figures 2D & 2E) of ) would be typo
Fig. 4E of X line should be MPP2A or HSC.
Author Response
Reviewer 1
Major points
- In the abstract, the author wrote, “To identify the contribution of senescent cells to aging of human hematopoiesis…..”
Corresponding to the suggestion of the Reviewer, we have changed the wording into: “To identify the differences between aged and young human hematopoiesis, we performed a direct comparison of aged and young human hematopoietic stem and progenitor cells (HSPCs)……" - line34, the author wrote, “A similar population in LT-HSC in mice could be validated.” there is no evidence of subpopulation imaged in mice data.
As suggested, we have changed this statement into:
“A population with similar senescence signature in the aged LT-HSCs was confirmed in murine aging model.“ - 1 (Paragraph 2.1) had no results. Please conclude the young vs aged GSEA analysis at the end of the paragraph.
We thank the reviewer for this proposal. We have now reorganized the subsections accordingly. In the “Results” section, we start with the subsection:
“2.1 GSEA between old and young subjects in HSPCs”, which begins with a description of the results of the single cell RNA sequencing studies and the genes identified, followed by the results of the GSEA analysis at the end of the paragraph. - In paragraph 2.3, they utilized 14 clusters and named each, but it wasn’t easy to follow considering the next paragraph, 2.5, of 6 clusters. How about changing the name of 2.3 (14 clusters) of HSC to HSPC?......
There were major formatting errors in Figure 2 in the version that the reviewers received. This has been corrected in the revised version. Figure 2A has now been enlarged such that the labeling of the 14 clusters is clear and legible. In addition, we have used the same color code for all the 14 clusters (explained in legend in Figure 2A) in all the figures 2A to 2E.
The reviewer is correct that Figures 2A to 2E deal with the clusters of HSPC, one of which, at the apex of the developmental trajectory, is the non-primed HSC cluster. Using additional bone marrow samples (from 15 instead of 12 human subjects) and a higher resolution in clustering the non-primed HSCs, we are able to identify 6 subclusters within the HSC cluster, i.e. multipotent progenitor 1 (MPP1), MPP2, MPP2A, MPP3, preMEP, and pre MLP. - Figure 2 is too small to follow the data. please show much larger figures….
We agree with Reviewer 1. As explained above, this mistake has now been corrected. - line263, “The P53-P21 pathway is closely linked to elevated expressions CDKN1A, and the P16 INK4A pathway to elevated CDKN2A and CDKN2B.”, line267 activated P53-P21 pathway and an elevated expression of CDKN1A”, line308”P53-P21 and activated CDKN1A”would be just redundant expression because CDKN1A is p21 and CDKN2A is p16. Rephrase or delete the sentence for the reader to easily follow.
We would like to thank the reviewer for pointing out this redundancy. We would like to emphasize that for human subjects, activated P53-P21, coupled with elevation of CDKN1A, is the underlying mechanism, and not p16 Ink4a and cdkn2a/cdkn2b as in murine models. We have now rephrased the statement:
“Our results have provided clear evidence that, in contrast to murine models, an activated P53-P21 pathway is the main mechanism for aging of HSC compartment in human subjects.” - line154, in Figures 2D & 2E) of ) would be typo
Thank you. This typo has been corrected. - 4E of X line should be MPP2A or HSC
Labeling of the X-axis has been added and revised to clarify the point.
Reviewer 2 Report
Comments and Suggestions for Authors
Hematopoietic stem and progenitor cells (HSPCs) play a key role in blood production. They are unique cells that can self-renew and differentiate into different types of blood cells. Understanding the regulatory mechanisms and therapeutic potential of these cells is fundamental to the field of regenerative medicine and cell therapy.
Currently, scientists are focusing on better understanding the molecular mechanisms regulating HSPC functions. Many growth factors and cytokines have been discovered that affect the proliferation and differentiation of these cells. In addition, research is being conducted on the potential use of HSPC in the treatment of blood diseases such as leukemia or anemia. With this in mind, I confirm that the article fits into current trends and will certainly be a source of information for other scientists.
Aging is associated with a number of molecular and cellular processes that affect the functioning of various systems, including the hematopoietic system. Studies conducted in animal models suggest that with aging, the number of HSPCs in the bone marrow may increase. However, paradoxically, despite this increased number, they show limited regenerative potential. This means that although there are more stem cells, their ability to regenerate tissue and produce new blood cells may be impaired compared to younger individuals.
Understanding the quantitative and functional changes in HSPCs in the context of aging is of crucial importance not only for the field of cell biology but also for regenerative medicine and cell therapy. As research on HSPCs advances, there is the potential to develop innovative therapies based on these cells. However, further knowledge of the mechanisms regulating HSPC functions in different phases of life is necessary to effectively exploit their therapeutic potential. These implications are discussed and analyzed by the authors in the introduction section of the article, providing a foundation for further considerations and analyses.
Then the authors move on to the results, and the research methodology and materials are placed in Chapter 4 - I would like this chapter, like other scientific articles, to appear before the results, creating a harmonious and orderly whole.
The results of the study include a comparison of the gene set enrichment (GSEA) of HSPC cells (CD34+ cells) from the bone marrow of the elderly and young. Scientific studies on the gene set of HSPC cells (hematopoietic stem cells) are crucial for understanding the biological processes related to regeneration and aging of the organism. Comparison of the gene set enrichment (GSEA) of HSPC cells collected from the bone marrow of the elderly and young can provide important information on the genetic changes occurring in these cells at different stages of life. Previous studies have focused on the analysis of gene expression in HSPC cells from the bone marrow of the elderly and young. These results indicate significant differences in the gene profiles between these two age groups. HSPC cells from the elderly often show changes in the expression of genes related to inflammation, aging and response to oxidative stress. In turn, HSPC cells from young people are characterized by the activity of genes related to growth, differentiation and resistance to stressors.
The discovery of the accumulation of two unique HSPC clusters in the bone marrow of older adults has numerous implications for the future of medicine and healthcare. First, it opens up new therapeutic possibilities for tissue regeneration and the promotion of the body's healing processes. Second, it allows for a better understanding of the aging process and the development of more precise treatment strategies related to it.
The conclusion from the conducted studies directs further research work towards finding alternative methods of identifying senescent cells. The dynamics of gene expression in HSCs remains an area of ​​intensive research, and the lack of surface markers may be an incentive to discover innovative research approaches. In the future, we can expect the development of new technologies that will allow for a better understanding and potential manipulation of stem cell aging processes, which may have a significant impact on the field of cell biology and regenerative medicine. As a result, research on HSCs and their aging remains a topic of great potential and complexity, requiring an interdisciplinary approach and constant exploration of new areas of knowledge.
In addition to the comment on the organization of the article, I would like to note that the manuscript also lacks a traditional summary that would factually summarize the results and outcomes obtained and the implications for the future. I believe that the article and the results contained in it are significant and worth publishing, but the disorganized structure does not make the work easier to read.
The authors used 68 scientific sources in their work, the selection of which I consider correct and accurate, and since these are current articles published quite recently, this only confirms that the topics discussed in this manuscript will attract many readers.

Author Response
Reviewer 2
We appreciate the encouraging and detailed comments by Reviewer 2 in the first four paragraphs of his critiques. “…..Understanding the quantitative and functional changes in HSPCs in the context of aging is of crucial importance not only for the field of cell biology but also for regenerative medicine and cell therapy…., further knowledge of the mechanisms regulating HSPC functions in different phases of life is necessary to effectively exploit their therapeutic potential. These implications are discussed and analyzed by the authors in the introduction section of the article, providing a foundation for further considerations and analyses.“
- Then the authors move on to the results, and the research methodology and materials are placed in Chapter 4 - I would like this chapter, like other scientific articles, to appear before the results, creating a harmonious and orderly whole.
Whereas I agree with Reviewer 2 that it might be better to place “Methods and Materials” before the section ”Results”, the Journal “IJMS” recommends this specific order for the various sections and we just have to comply with the recommendation. - The results of the study include a comparison of the gene set enrichment (GSEA) of HSPC cells (CD34+ cells) from the bone marrow of the elderly and young. Scientific studies on the gene set of HSPC cells (hematopoietic stem cells) are crucial for understanding the biological processes related to regeneration and aging of the organism. Comparison of the gene set enrichment (GSEA) of HSPC cells collected from the bone marrow of the elderly and young can provide important information on the genetic changes occurring in these cells at different stages of life. Previous studies have focused on the analysis of gene expression in HSPC cells from the bone marrow of the elderly and young. These results indicate significant differences in the gene profiles between these two age groups. HSPC cells from the elderly often show changes in the expression of genes related to inflammation, aging and response to oxidative stress. In turn, HSPC cells from young people are characterized by the activity of genes related to growth, differentiation and resistance to stressors.
We deeply appreciate these encouraging comments expressed by Reviewer 2 and completely agree with her/him. - The discovery of the accumulation of two unique HSPC clusters in the bone marrow of older adults has numerous implications for the future of medicine and healthcare. First, it opens up new therapeutic possibilities for tissue regeneration and the promotion of the body's healing processes. Second, it allows for a better understanding of the aging process and the development of more precise treatment strategies related to it.
We thank Reviewer 2 for this recognition of the significance of our manuscript. The discovery of two unique HSPC clusters – MPP2A as a unique subcluster within aged HSCs, and Ly4 as another cluster further downstream in the lymphoid developmental trajectory will open up new avenues for regenerative medicine and for a better understanding of the aging process in human hematopoiesis. - The conclusion from the conducted studies directs further research work towards finding alternative methods of identifying senescent cells. The dynamics of gene expression in HSCs remains an area of ​​intensive research, and the lack of surface markers may be an incentive to discover innovative research approaches. In the future, we can expect the development of new technologies that will allow for a better understanding and potential manipulation of stem cell aging processes, which may have a significant impact on the field of cell biology and regenerative medicine. As a result, research on HSCs and their aging remains a topic of great potential and complexity, requiring an interdisciplinary approach and constant exploration of new areas of knowledge.
We agree with Reviewer 2. We would like to add that the identification of an analogous senescent population in murine model is the best we can accomplish to prove the role of senescent population in the aging human HSC compartment. Elimination or reduction of this senescent population has rejuvenated hematopoiesis in mouse models (Ref. 22 & 23, Baker et al; Ref. 33 Chang et al. in the manuscript). - In addition to the comment on the organization of the article, I would like to note that the manuscript also lacks a traditional summary that would factually summarize the results and outcomes obtained and the implications for the future. I believe that the article and the results contained in it are significant and worth publishing, but the disorganized structure does not make the work easier to read.
I agree that the previous version might appear disorganized, especially with the formatting errors in presenting Figures 2A to 2E, and with the misplacement of the legend for Figure 2. All these formatting errors have been corrected. We have also added a “5. Conclusions” section to summarize the results, their implications, and the prospects for the future. - The authors used 68 scientific sources in their work, the selection of which I consider correct and accurate, and since these are current articles published quite recently, this only confirms that the topics discussed in this manuscript will attract many readers.
Thank you. We agree with your opinion and have left the reference list as it was.
Round 2
Reviewer 1 Report
Comments and Suggestions for Authors
I improved the revised version based on the reviewer's suggestions.